# A Study of Urban Planning in Tsunami-Prone Areas of Sri Lanka

**U. T. G. Perera [1,*], Chandula De Zoysa [2], A. A. S. E. Abeysinghe [3], Richard Haigh [1], Dilanthi Amaratunga [1] and Ranjith Dissanayake [3]**

1 Global Disaster Resilience Centre, Department of Biological and Geographical Sciences, School of Applied Sciences, University of Huddersfield, Queensgate, Huddersfield HD1 3DH, UK
2 Department of Civil Engineering, University of Moratuwa, Bandaranayake Mawatha, Moratuwa 10400, Sri Lanka
3 Department of Civil Engineering, University of Peradeniya, Galaha 20400, Sri Lanka
* Correspondence: thisara.uduwarage@hud.ac.uk

**Abstract:** Tsunamis pose significant challenges for disaster reduction efforts due to the multi-hazard, cascading nature of these events, including a range of different potential triggering and consequential hazards. Although infrequent, they have the potential to cause devastating human and economic losses. Effective urban planning has been recognised as an important strategy for reducing disaster risk in cities. However, there have been limited studies on urban planning for tsunami-prone areas, and there have been wide ranging strategies adopted globally. This is an international study aimed at exploring the status of urban planning in tsunami areas and better understanding potential urban planning strategies to reduce disaster risk in coastal regions. Drawing upon the work of an international collaborative research team, in this article, we present the findings of a systematic review of the urban planning literature. Using the PRISMA guidelines, 56 papers were selected, and three guiding questions informed the review. Further empirical investigations were carried out in Sri Lanka by a local research team, including twelve semi-structured interviews with representatives from agencies in urban planning, construction, and disaster management, and a focus group representing town and country planning, architecture, structural engineering, disaster management, landscape and geospatial planning, building services, green buildings and infrastructure and environmental management fields. The combined analysis reveals insights into the characteristics of the literature, as well as the nature of existing strategies for urban planning in tsunami-prone areas, grouped into six broad themes: community participation, spatial planning, soft and hard engineering;,evacuation planning, and resilience thinking. The findings also reveal limitations in existing strategies, including their failure to address multi-hazard threats and systemic risk, as well as inadequate community participation, and limited access to timely disaster risk information. The findings are used to inform an initial model of urban planning strategies in tsunami-prone areas that can be used before a hazard event occurs, during and in the immediate response to a hazard event, and during recovery and reconstruction following a disaster.

**Keywords:** urban planning; tsunami; Sri Lanka; multi-hazard

## 1. Introduction

Tsunamis pose significant challenges for disaster reduction efforts due to the multi-hazard, cascading nature of these events, including a range of different potential triggering and consequential hazards. Although infrequent, they have the potential to cause devastating human and economic losses.

Significant international and national efforts have gone into developing tsunami early warning systems, especially for the Pacific and Indian Ocean regions. However, recent tsunami events have illustrated the limitations of these systems, especially for near field events that may offer a short mobilisation period for a tsunami threat to be detected and warning information to be issued to people at risk, and thereby, enable effective evacuation and response efforts.

Effective urban planning has been recognised as an important strategy for reducing disaster risk in cities. However, there have been limited studies on urban planning for tsunami-prone areas and there have been wide ranging strategies adopted globally. Many also fail to consider the complexity of multi-hazard threats.

An international study was undertaken to explore the status of urban planning in tsunami-prone areas in order to better understand potential strategies to reduce disaster risk in these coastal regions. Drawing upon the work of a collaborative research team, in this article, we present the findings of a systematic review of the international urban planning literature on tsunami-prone areas and a detailed qualitative study in Sri Lanka.

Sri Lanka is a member state of the Indian Ocean Tsunami Warning and Mitigation System (IOTWMS) and was severely impacted by the 2004 Indian Ocean Tsunami. Despite only experiencing a single devastating tsunami event in modern history, a tsunami has been identified as the highest risk index among all hazard types in the country (8.9 out of 10) [1]. As highlighted by Haigh et al. [2], prior to 2004, there was little in the way of formal preparedness for tsunami early warning in the country and there was a lack of experience among the people, increasing their vulnerability. After 2004, there has been widespread recognition of a need for Sri Lanka to actively engage in disaster risk reduction efforts because of its vulnerability to future tsunami threats [3]. These investments in mitigation and preparedness for tsunami make Sri Lanka a useful focus for further detailed enquiry and to examine the extent to which urban planning features among those efforts. Three guiding research questions were used to scope the data collection and analysis.

### 1.1. Coastal Cities and Disaster Risk

The world has faced an era of exponential growth in its urban population. Cities are already home to more than half of the world's population, and are centres of economic growth and innovation [4]. Many of these cities are also situated in hazard-prone areas—along coasts or on floodplains, on top of or near seismic faults, in the shadow of volcanoes, and in areas prone to tropical cyclones and extreme storms. These natural hazards tend to exert even greater impacts since cities concentrate people and economic assets, including large-scale infrastructure and properties [4]. Specific threats include rapid-onset hazards such as major cyclones accompanied by high winds, waves and surges, or tsunamis. They are also threatened by slow-onset hazards, including erosion and gradual inundation [5]. The effects of climate change can be especially devastating to vulnerable coastal areas, including the function and structure of their ecosystems. Sea level rise changes the shape of coastlines, and contributes to the risk of coastal erosion, flooding, and salt-water intrusion. The UNFCC [6] estimates that more than 600 million people (around 10 per cent of the world's population) live in coastal areas that are less than 10 m above sea level, and these occupy 7% of the Earth's land area.

### 1.2. Tsunami and Its Complexity as a Cascading Disaster

Tsunamis are typically triggered by earthquakes, but can also be triggered by other hazards, including volcanic eruptions, submarine landslides, and by onshore landslides in which large volumes of debris fall into the water. Although less frequent than most other natural hazards, tsunamis have the potential to cause devastating property damage and loss of life. The International Tsunami Information Centre reports that over the last twenty years, deadly tsunamis have been triggered near Chile (2007, 2010), Haiti (2010), Indonesia (2004, 2005, 2006, 2010, 2018 (two separate events)), Japan (2011), Peru (2001), Samoa-American Samoa-Tonga (2009, 2022), and the Solomons (2007). Although such

devastating events have been rare, some recent studies have suggested that even minor sea level rise will increase the risk of tsunamis for coastal communities [7]. Similarly, the growth of urban populations in coastal regions is likely to increase exposure of people and assets to the threat posed by tsunamis.

The 2004 Indian Ocean Tsunami, which impacted twelve countries, is one of the worst disasters in recorded human history. The 2011 Tohoku Earthquake and Tsunami event, often referred to as the Great East Japan Earthquake and Tsunami, caused the sea water to reach more than 5 km inland in some coastal locations, reaching a height of 19.5 m above sea level. In both events, larger numbers of people lost their lives and were displaced, losing their homes and property. [8–10].

In late 2018, Indonesia was hit by two further, destructive tsunamis, which as reported by IOC [10], challenged the traditional understanding of tsunami hazard, warning, and response mechanisms. At an International Symposium on the Lessons Learnt from the 2018 Tsunamis in Palu and Sunda Strait [11], experts identified a need for a critical dialogue on the future direction of the Indian Ocean Tsunami Warning and Mitigation System (IOTWMS), especially for events other than tectonic origins and with short warning times, referred to as local source or near field tsunami. They concluded that the national warnings issued in Indonesia within 5 min of the earthquake, were of limited practical use for the Palu tsunami, especially in coastal areas where tsunami waves arrived in less than 3 min. They also determined that current early warning systems are most effective for tsunamis generated by subduction zone earthquakes but have limitations to handle atypical (landslide and volcanic) and/or near-field tsunamis.

Tsunamis such as the one in Palu are often associated with multi-hazard threats with cascading effects. Goda et al. [12] explained that the cascading effects of the 2018 Sulawesi earthquake were very complex and varied widely in space, which triggered secondary hazards, such as a tsunami, liquefaction, submarine landslides, and massive mudflows in coastal areas. Muhari [13] added that such cascading effects could increase in progression over time and generate unexpected secondary events of strong impact across the coastal regions.

The 2011 Tohoku Earthquake and Tsunami event also illustrated the potential devastation caused by cascading effects of earthquakes and tsunamis [14], in this case, damage to the local nuclear power plant caused extensive radioactive contamination. It also caused widespread damage to critical infrastructure, including transport such as roads, 71 bridges, and 26 parts of the railway system, and lifeline infrastructure such as electricity, water supply, sewage systems, and gas lines.

These previous disasters illustrate the complexity facing tsunami early warning and preparedness efforts, which must overcome the dilemma of time verses uncertainty due to short response times, limitations of technology, and currently available scientific knowledge. There has been considerable progress and improvement in the IOTWMS that has been developed since the devastating 2004 Indian Ocean Tsunami. However, the UN coordinating agency for the IOTWMS has formally recognised that much remains to be done to ensure dissemination of effective warnings and to prepare communities to act upon them [11,15]. Most recently, recommendations from the Lessons Learnt Symposium [11] stressed a need to address the lack of proper evacuation plans and related infrastructure during a tsunami emergency, as well as poorly implemented/ineffective spatial planning and policy-related issues.

Poorly planned urban environments, weak urban governance, and old and fragile infrastructure can all increase pressure on the urban environment and trigger exposure to these cascading effects. They are associated more with the magnitude of vulnerability of the coastal regions than with that of the hazard. As highlighted by Rus [16], the need for maintenance and upkeep of these cities makes safety measures for their citizens crucial in these situations. Therefore, urban planning has been recognised as a key factor to build the resilience of urban coastal communities in the face of increasing hazards and disaster risk.

*1.3. Urban Planning and Disaster Risk*

Urban planning encompasses the preparation of plans for and the regulation and management of towns, cities, and metropolitan regions. It attempts to organize socio-spatial relations across different scales of government and governance, and is concerned with the social, economic, and environmental consequences of delineating spatial boundaries and influencing spatial distributions of resources [17]. In addition, it focuses on attempting to improve the plans, functions, and management of cities and areas, which has a clear role to play in disaster mitigation [18].

Urban planning can serve as a key tool for limiting the disastrous effects of hazards and for enhancing community resilience. This has been emphasized in recent international agreements, i.e., the Hyogo Framework for Action (2005) [19], the Sendai Framework for Disaster Risk Reduction (2015) [20], and the 2030 Agenda for Sustainable Development (2015) [21], which highlight the importance of incorporating disaster risk reduction and resilience enhancement into urban and regional planning processes [22].

Alan [23] also highlighted the potential to reduce disaster risk through proper urban planning approaches, where understandings of risk are increasingly being integrated within the wider processes of urban planning. For example, proper planning allows cities to control land use and occupation, which is a crucial factor in minimising exposure to a tsunami and flooding. Many recent studies have also highlighted the importance of land use planning in coastal regions as a crucial factor for urban resilience [16,24,25].

Some countries have used a variety of urban planning instruments to monitor land use and minimize the effects of specific threats such as tsunamis. These include resettlement, the development of no-build zones, land use restrictions, and tsunami-safe building standards, as well as the preservation of natural buffers including forests, compacted dunes, and wetland areas [26]. Furthermore, Eisenman [27] highlighted the importance of non-statutory policies such as evacuation, mandatory insurance, emergency, and the promotion of social organizations on community resilience, which can be promoted through effective urban planning.

Although some of these measures may be suitable for dealing with tsunamis, they may not be appropriate measures for other types of hazards or threats, for instance, seismic activity, severe storms, or landslides. Bosher [28] pointed out that multi-hazard/threat assessments should be undertaken, and any risk reduction options should be proportionately considered alongside any other hazards or threats that have been identified. Despite this recognition of the need, Ma et al. [29] found that the vulnerability of a city or area to hazards was not being adequately evaluated, especially in developing countries and even where geo-hazards were likely to occur from time to time due to the poor physical environment and strong human activities. Sengezer and Koç [30] also revealed that a high risk of exposure to multiple hazards in such areas is associated with considerable deficiencies in the social and political spheres. This includes a lack of professional training programmes, poor access to technology, lack of experience of organisations, inefficient professional chambers, and inadequate control mechanisms. These poor mechanisms have the potential to do serious harm to people's lives and property in coastal regions of the world [31].

The "new normal" that is expected to emerge as the world recovers from COVID-19 is also likely to further change the complexity of disaster risk, as urban areas change to reflect societal changes, such as remote working [32,33]. Urban planning will need to evolve so that it can continue to contribute to mitigation and preparedness efforts. A better understanding of the trends and dynamics of pandemics, and their impact on urban planning for coastal cities, response, and adaptation measures, will be vital in tackling this dynamic situation.

Although a large body of research has been published on various strategies related to urban planning in coastal areas, there appears to be a lack of consensus or best practice around what measures to adopt in tsunami-prone areas. The combined effects and cascading effects of hazards are also not addressed in most of the existing research.

## 2. Materials and Methods

This study aims to explore the state of the art in urban planning for tsunami-prone areas and to better understand good practices and gaps in the knowledge base. Through a series of meetings among an international research team from Sri Lanka and the UK, the following research questions were developed to guide the study:

**RQ1** What are the characteristics of the literature on urban planning in tsunami-prone areas?

**RQ2** What are the existing strategies and approaches for urban planning in tsunami-prone areas?

**RQ3** What are the main gaps and limitations in those strategies?

A systematic review of the literature was carried out by the UK research team on urban planning strategies in tsunami-prone areas. The Preferred Reporting Items of Systematic Reviews and Meta-Analysis guidelines (PRISMA) were used to prepare and facilitate transparent reporting of the systematic review [34]. This guideline enables readers to assess the quality and validity of the reporting. After the literature search, a frequency analysis was undertaken to identify and categorise the applied methods from literature to better frame the urban planning strategies.

The search terms "(Urban Planning* AND (Tsunami) AND/OR (Multi Hazards)" were used in Summon and University of Huddersfield Library Services, Scopus, and Web of Science. These research services were used to reach a wide range of research across multiple databases. These publication platforms reliably search across publishers and are not bias towards journals published by any one company. In total, the search methods identified 1918 records. The year range of 2000–2020 was applied to focus on the considerable body of literature that had emerged since the 2004 Indian Ocean Tsunami and to ensure that it reflected the most recent studies under the urban planning literature. After searching all the databases, the studies were added to EndNote software and the duplicate studies were deleted. Further, the manual search method was applied in the EndNote software to find duplicates. Then, the titles and summaries were reviewed to find relevant studies. Subsequently, experienced scholars specializing in the field independently reviewed the full text of the articles and applied the following inclusion criteria:

1.  Research papers, book chapters, journal articles are considered;
2.  Research should include the term/terms in the title abstract or the keywords;
3.  It should be in English;
4.  Research should be related to one or more domains of urban planning strategies in tsunami-prone areas.

The following exclusion criteria were also applied:

1.  Urban planning is not a main focus;
2.  Does not include tsunamis or other coastal hazards;
3.  Does not include urban planning strategies or mechanisms;
4.  Does not address research questions of the study.

After applying the inclusion and exclusion criteria, 56 academic research papers (Appendix A Table A1) were selected and reviewed to address the research questions (Figure 1).

In addition to the systematic review, empirical investigations were planned and carried out in Sri Lanka by a local research team but using a methodological approach jointly developed by the UK and Sri Lanka team. These contemporary insights into current urban planning for tsunami-prone areas in Sri Lanka, including its current strategies, foci, and limitations, complement and contextualize the findings from the systematic review, which provides much broader insights.

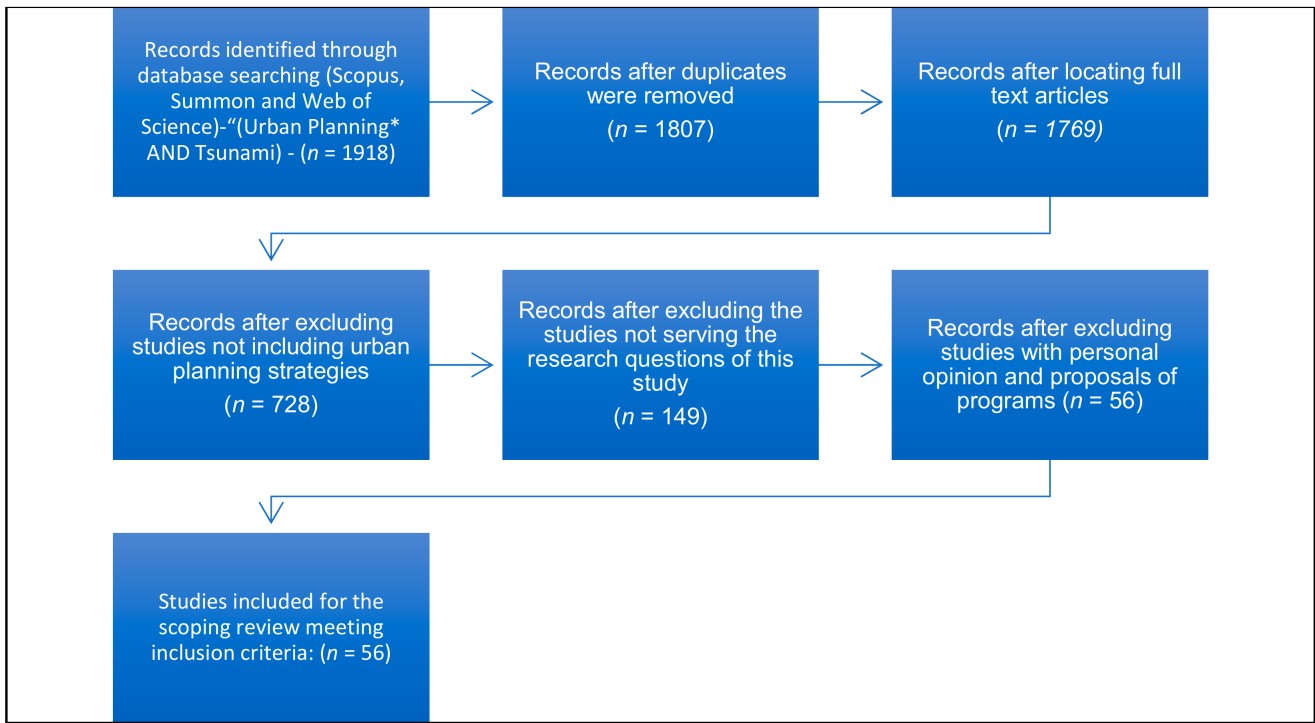

**Figure 1.** Flow diagram of the systematic literature review.

The empirical investigations involved semi-structured expert interviews and a focus group to investigate the existing strategies and approaches for urban planning in tsunami-prone areas of Sri Lanka, systemic risk and cascading impacts, and the constraints and pathways to mainstreaming disaster risk reduction into urban planning at the national to local levels. This dual approach combined the benefits afforded by the detail and individual contexts from interviews, as well as the wider range of ideas and greater "surface" data from a focus group [35]. It also enabled convergence of the current status, good practices, and gaps across focus groups and individual interviews, which enhanced the trustworthiness of the findings.

Twelve interviews were conducted in Sri Lanka with representatives from agencies in urban planning, construction, and disaster management (Table 1). These interviews were an opportunity to explore beyond the literature and allow additional concepts, themes, and areas of analysis to be discovered. Further, the focus group in Sri Lanka was used to examine further and validate the findings from the interviews, while also providing an opportunity to clarify issues and ensure nothing significant was overlooked. Six experts took part in a one and half hour focus group discussion, representing town and country planning, architecture, structural engineering, disaster management, landscape and geospatial planning, building services, green buildings and infrastructure, and environmental management fields. An audio recording of the focus group discussion was analysed using content analysis.

Data collection for the systematic review and interviews/focus group was conducted concurrently and largely independently. Then, both strands of research were combined during data analysis and, primarily, interpretation. Findings from the systematic review were cross-referenced, triangulated, and contextualized with expert observations to gain a deeper understanding of urban planning strategies in tsunami-prone areas within the theoretical discourse, research gaps, and potential discrepancies, with observations from practice.

**Table 1.** Interview respondent expertise.

| Expert Area | Code |
|---|---|
| Architecture, Town Planning | A1 |
| Architecture, Town Planning | A2 |
| Architecture, Town Planning | A3 |
| Civil Engineering | C1 |
| Civil Engineering, Disaster Management | C2 |
| Civil Engineering, Disaster Management, Sustainable Built Environment | C3 |
| Civil Engineering | C4 |
| Disaster Management | D1 |
| Disaster Management | D2 |
| Green Building Consultant, Town Planning | G1 |
| Town and Country Planning | T1 |
| Town and City Planning | T2 |

*Data Analysis*

A frequency analysis was conducted to record the major concerns and dimensions for urban planning strategies found through the systematic review, as well as the aspects of the urban planning applied in the literature. In the review, the data was organised by the following categories: case information, characteristics of urban planning, dimensions of urban planning, and applied methods. The case information included source, types of hazard described in the study (topic), the spatial scale, and the type of territory where the assessment methods applied. The spatial scale was further categorized by the size at which the assessment was operated. It ranged from the community, district, city, region, country to the global scale. Further analysis for the systematic review included expert interviews and a focus group that were developed based on a thematic extraction, where key themes of each research and interviews were extracted. Then, the themes were classified based on the research questions, and these key findings of the classification are highlighted in the results and discussion section. Based on the above analysis, the key strategies were derived to form a conceptual framework related to urban planning in tsunami-prone areas.

## 3. Results

### 3.1. Characteristics of the Literature

3.1.1. Location of the Research

Figure 2 illustrates the geographical location of the selected research studies in the systematic review. The Indian Ocean tsunami in 2004 appears to have prompted a substantial body of studies related to urban planning, with several countries in the region strongly represented. Indonesia, which experienced the highest death toll in the disaster, has been the most frequent target of studies, with 11 of the 56 addressing the country. Indonesia has also experienced very high levels of population and economic growth, and has urbanized areas spread across 6000 inhabited islands, and thereby, faces major challenges in terms of urban development. Sri Lanka ($n = 7$), the country that experienced the second highest death toll from the 2004 tsunami, also features prominently in the studies, while there are also a smaller number of studies in other affected countries, including India ($n = 4$), Bangladesh ($n = 1$), Malaysia ($n = 1$), Thailand ($n = 1$), and the Maldives ($n = 1$). Several Pacific Ocean countries also feature prominently, including Japan ($n = 5$), which suffered high human and economic losses from the 2011 Tohoku Earthquake and Tsunami, and Chile ($n = 7$), which experienced a tsunami following a magnitude 8.8 earthquake in 2010.

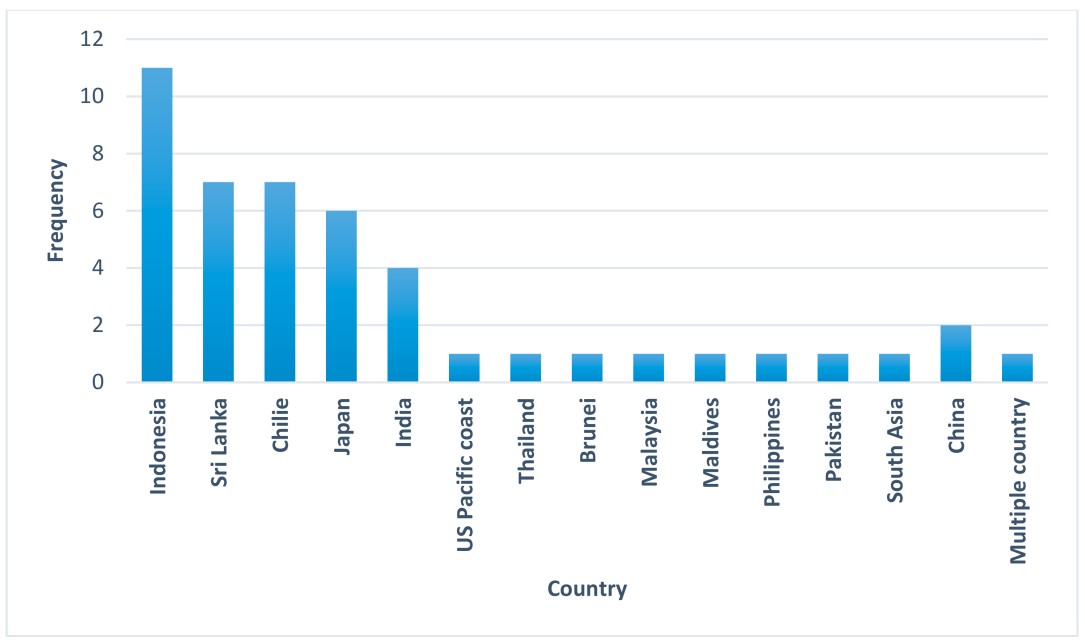

**Figure 2.** Study area of the selected sources.

### 3.1.2. Spatial Scale

Figure 3 illustrates the spatial scale of the reviewed sources in the systematic review. City level and country level scales were the prominent focus in most of the research. These two scales covered more than half of the reviewed sources (60%). Another characteristic from the literature included in this study is that the majority of research has been concentrated on larger cities (such as Jakarta, Aceh, and Padang), whereas the latter (smaller cities and towns) have been far less explored and therefore understood. This is significant because smaller cities and towns are likely to have different features and urban planning issues than bigger, major cities. Smaller cities and villages, for example, are likely to have less decentralised power and resources than capital cities, which may have an influence on local action in urban planning response to catastrophe risk [36].

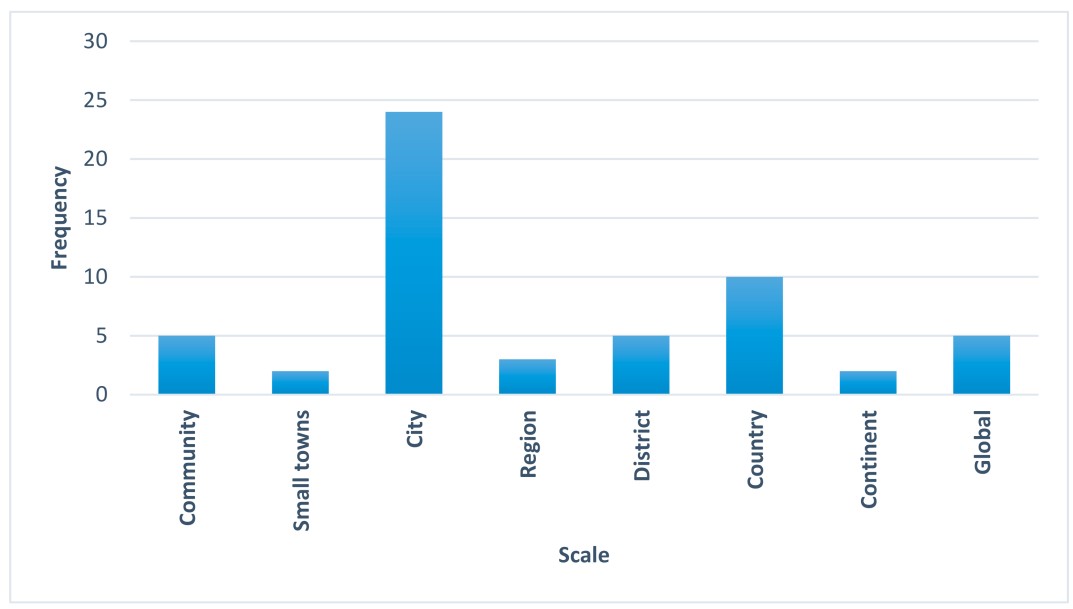

**Figure 3.** Spatial scale of the selected sources.

### 3.1.3. Year of Publication

Figure 4 sets out the frequency of included studies against the year of publication. A substantial increase in the number of publications can be observed from 2013. In short, the majority of included studies were published in the last six to seven years. This could be due to an increase in awareness about urban planning and their role in tackling disaster risk, or an overall increase in disaster risk and major losses linked to natural hazards. It could also be an indication that this is an area of research that is growing in interest, with more work being published. Alternatively, it may also reflect an increase in the number of disaster risk related journals and overall publication volumes in recent years. Regardless, the implications of that this is an area of increasing interest, and it is very likely that in the next few years there will be more relevant literature in this area.

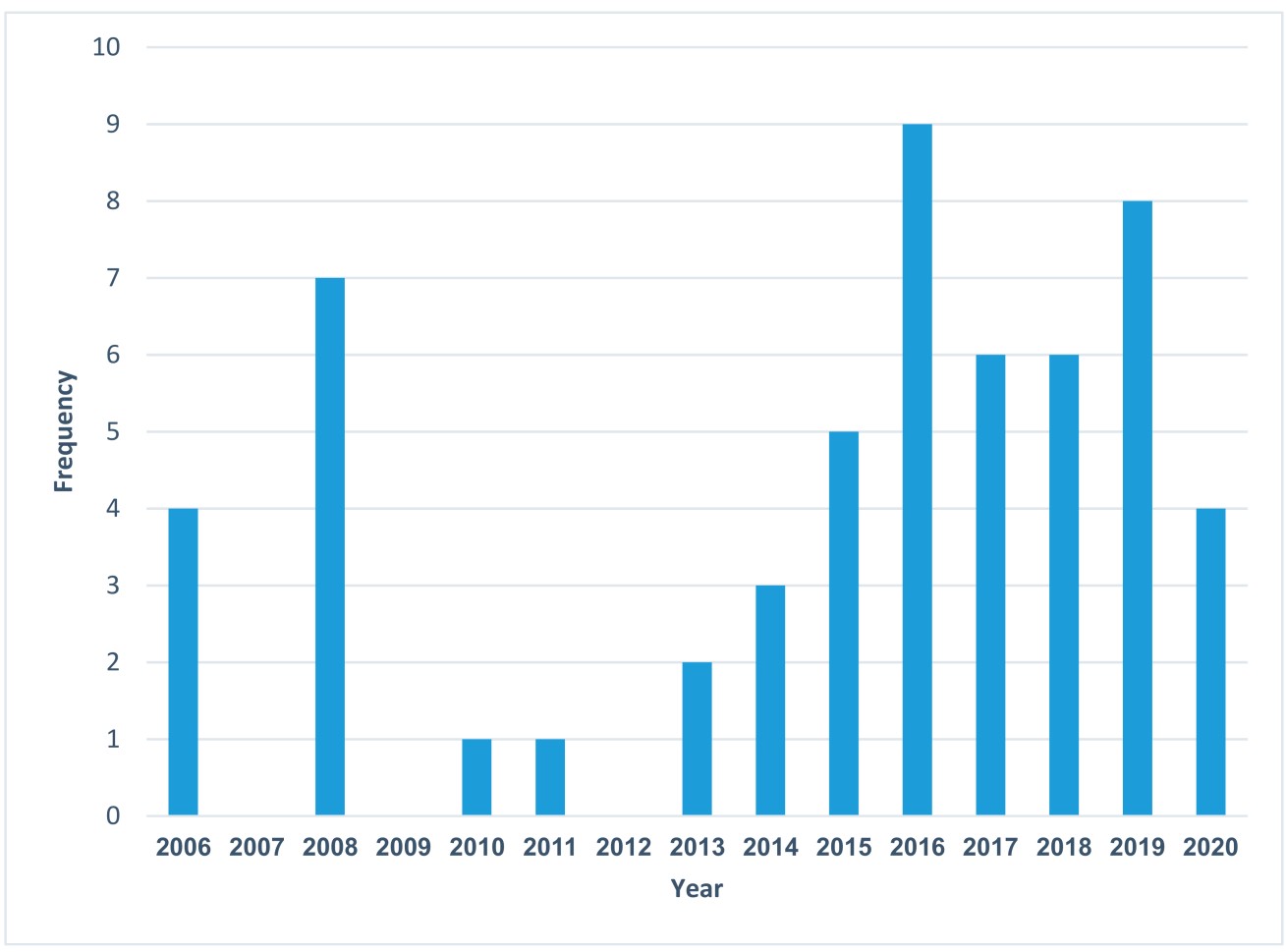

**Figure 4.** Published year of the selected sources.

### 3.1.4. Research Methods

In terms of research methodology and as represented in Figure 5, a wide range of different approaches were adopted across the studies. A general observation was the predominance of studies utilising qualitative approaches, most frequently based on perception data such as a limited number of purposely sampled interviews, literature, or secondary data. Despite all selected articles being in academic peer-reviewed journals, several had no clearly articulated method. Even those that did, tended not to describe data collection techniques at length, and many did not describe detailed data analysis approaches. An in-depth analysis of the quality of quality of evidence, such as weight

of evidence, was not carried out. However, it was observed that many used purposive samplings, had small sample sizes, or did not specify in sufficient detail.

The lack of common methodological approaches across studies, as well as the lack of detail in the presentation of research methods, makes direct comparisons across studies more difficult. It also does not lend itself to research synthesis, which could otherwise be used to inform urban planning in different contexts.

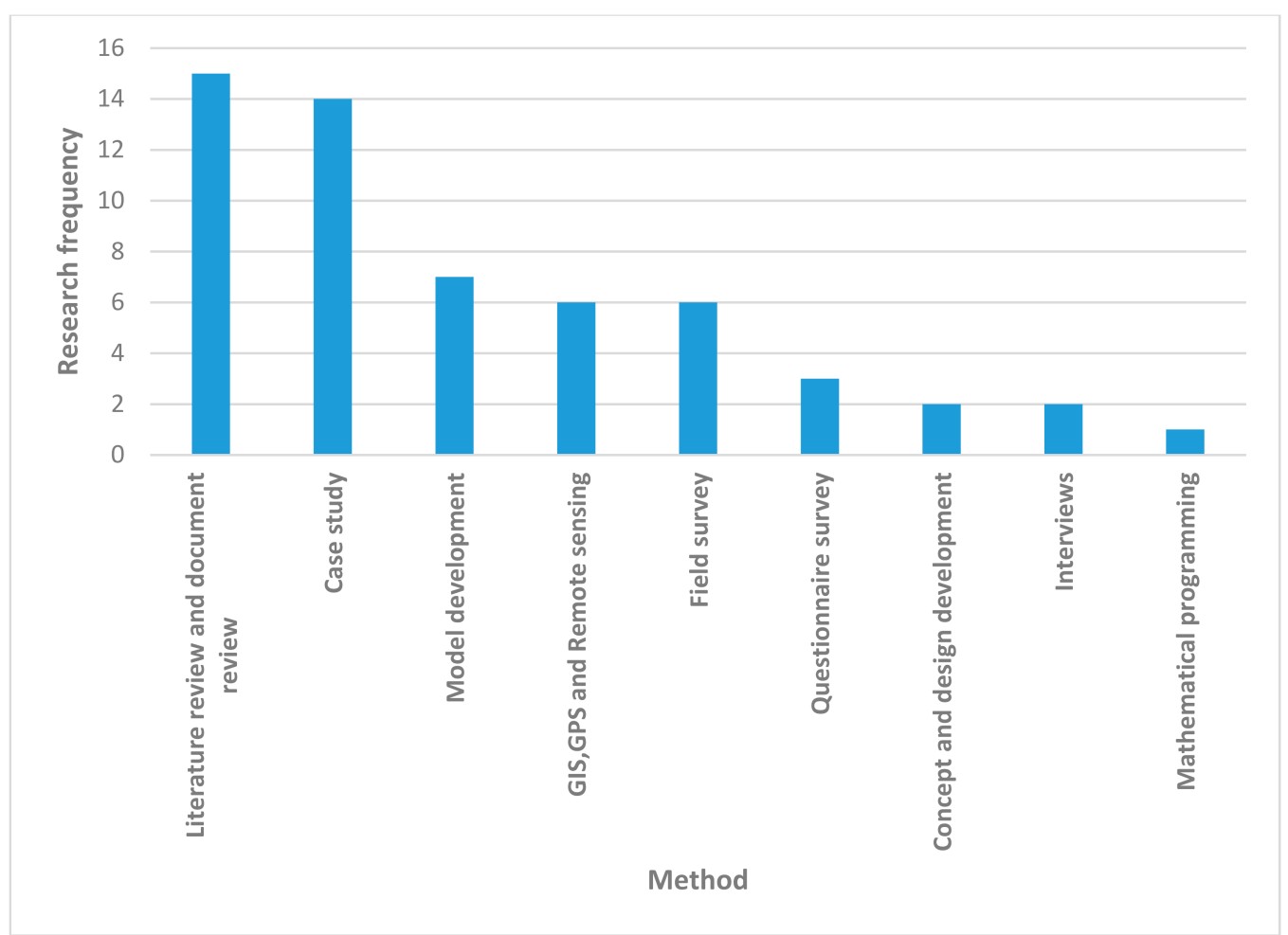

**Figure 5.** Research methods of the selected sources.

3.1.5. Other Hazards in Tsunami-Prone Areas

As the main focus of the study, tsunami hazards feature prominently in the selected studies, with over 30 either completely or partially investigating tsunami risk in coastal cities. As illustrated in Figure 6, other hazards in the selected papers include floods (*n* = 7), pandemics (*n* = 2), earthquakes (*n* = 1), and cyclones (*n* = 1). Figure 4 sets out the frequencies of studies included in this systematic review focusing on different urban hazards. Other hazards, such as coastal storms, wildfires, and landslides were not discussed in the urban planning literature. This resonates with Murray [36] who also noted in their review that many of the studies on urban risk focused on floods, earthquakes and tsunamis, whereas other hazards are underrepresented.

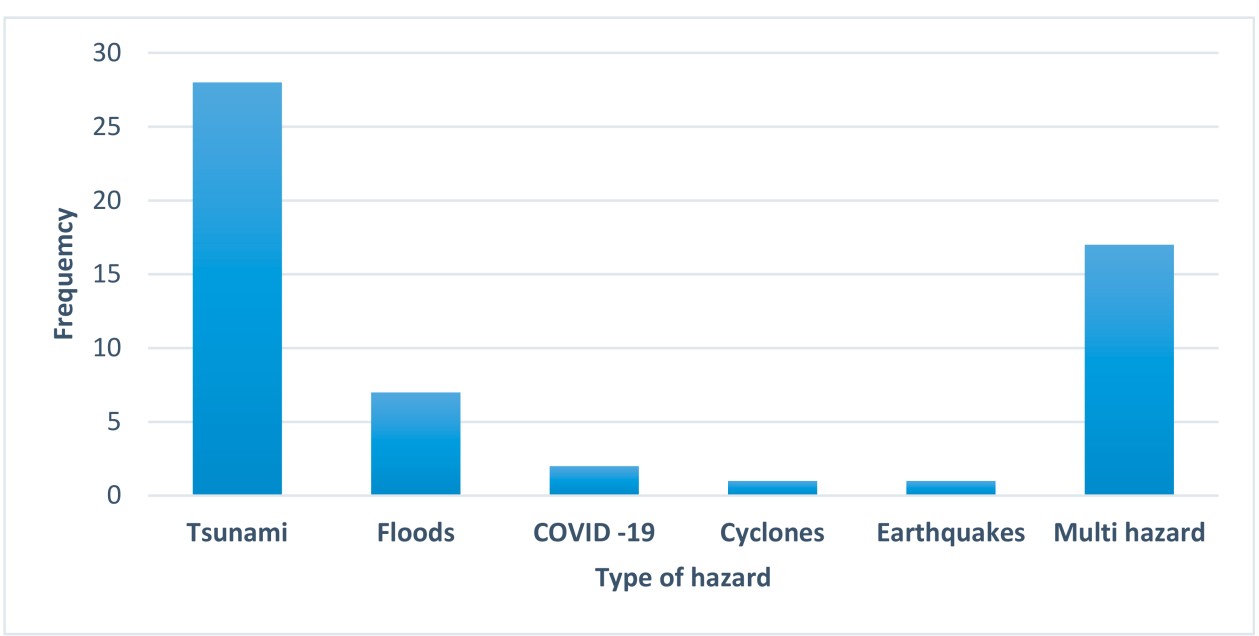

**Figure 6.** Frequency of hazards in tsunami-prone areas in selected sources.

Among the other hazards in tsunami-prone areas, floods in coastal regions were frequently highlighted as the prominent hazard, especially in South and South East Asia [37,38]. Dhiman [37] revealed that the risks posed by floods in coastal cities were primarily due to (1) the riverine flooding and extreme precipitation; (2) the effect of cyclonic storms, tsunamis, and tidal inundation; (3) failure in urban planning and unsustainable land use practices. The impact of earthquakes is also discussed in some of the papers combined with tsunamis, which pose a great threat to coastal cities as a stand-alone event or a tsunamigenic event. The 2004 Indian Ocean Tsunami, which was triggered by an earthquake with an epicentre off the west coast of Northern Sumatra, Indonesia, affected many coastal cities in Southeast and South Asia. Most of the reviewed papers highlighted the impact of the event, which also prompted planners to develop urban planning strategies to tackle similar events in the future [39,40]. Among the selected urban planning literature, there were very few studies on biological hazards, such as the COVID 19 pandemic and Ebola outbreak (*n* = 2). Due to the wide reach and devastating impact of the COVID-19 pandemic, it is reasonable to expect that more studies that address biological hazards will emerge soon.

A substantial number of studies identified in this review were multi-hazard (*n* = 17), focusing on more than one hazard and often the interactions between hazards. For example, tsunamis are often a significant secondary hazard following an earthquake, especially in coastal cities in countries such as Chile, Indonesia, and Japan, which are located in the most seismically prone region of the world, the Pacific "ring of fire" [41]. Similarly, coastal cyclones are also liable to cause floods where urban settlements spread in an unplanned way along the coast, such as found in Mumbai and Kolkata [42]. Barría [43] also highlighted that flood risks were often coupled with cyclones, storm surges, tides, and tsunamis across different geographical locations in a multi-hazard scenario. Most of the studies highlighted the importance of integrating multi-hazard scenarios in urban planning, and thereby, to provide a proper estimation of the risk level in the regions. Doerner [44] cautioned that there was a high probability for a combination impact of two or more hazards, such as tsunamis and earthquakes or floods and landslides in the same time frame. Similarly, Goda [12] stressed the high probability of cascading effects due to multi-hazard scenarios.

*3.2. Existing Strategies for Urban Planning in Tsunami-Prone Areas*

As illustrated in Figure 7, the literature describes a range of urban planning strategies that have been implemented in tsunami-prone areas that could contribute to disaster risk reduction. The most discussed strategies include evacuation planning (13), community participation (12), and land use planning (11). Less frequently discussed are resilience thinking (6); soft engineering (4); physical and social (3); sustainable housing, and physical and health planning (2 each); and hard engineering, urban physical growth planning, and urban governance (1 each).

Evacuation planning is the most frequently discussed strategy in the reviewed sources (*n* = 13). Most of the reviewed papers highlighted the importance of evacuation planning as an urban planning strategy for disaster risk reduction in coastal regions. Further, these reviewed sources have applied evacuation planning strategies in connection with a range of coastal hazards, including tsunamis, earthquakes, floods, and cyclones. Yossyafra [45] also contended that many governments have paid more attention to evacuation planning for their disaster risk reduction efforts.

Among the studies on evacuation planning, a number of different approaches emerge. An improved road infrastructure system (*n* = 6) is frequently discussed as a way to improve evacuation planning. Safe routes development with emergency shelters are also commonly discussed (*n* = 6). Public open spaces (*n* = 2), escape hill development (*n* = 2), and vertical evacuation (*n* = 2) are some of the other evacuation planning approaches that are suggested as part of urban planning.

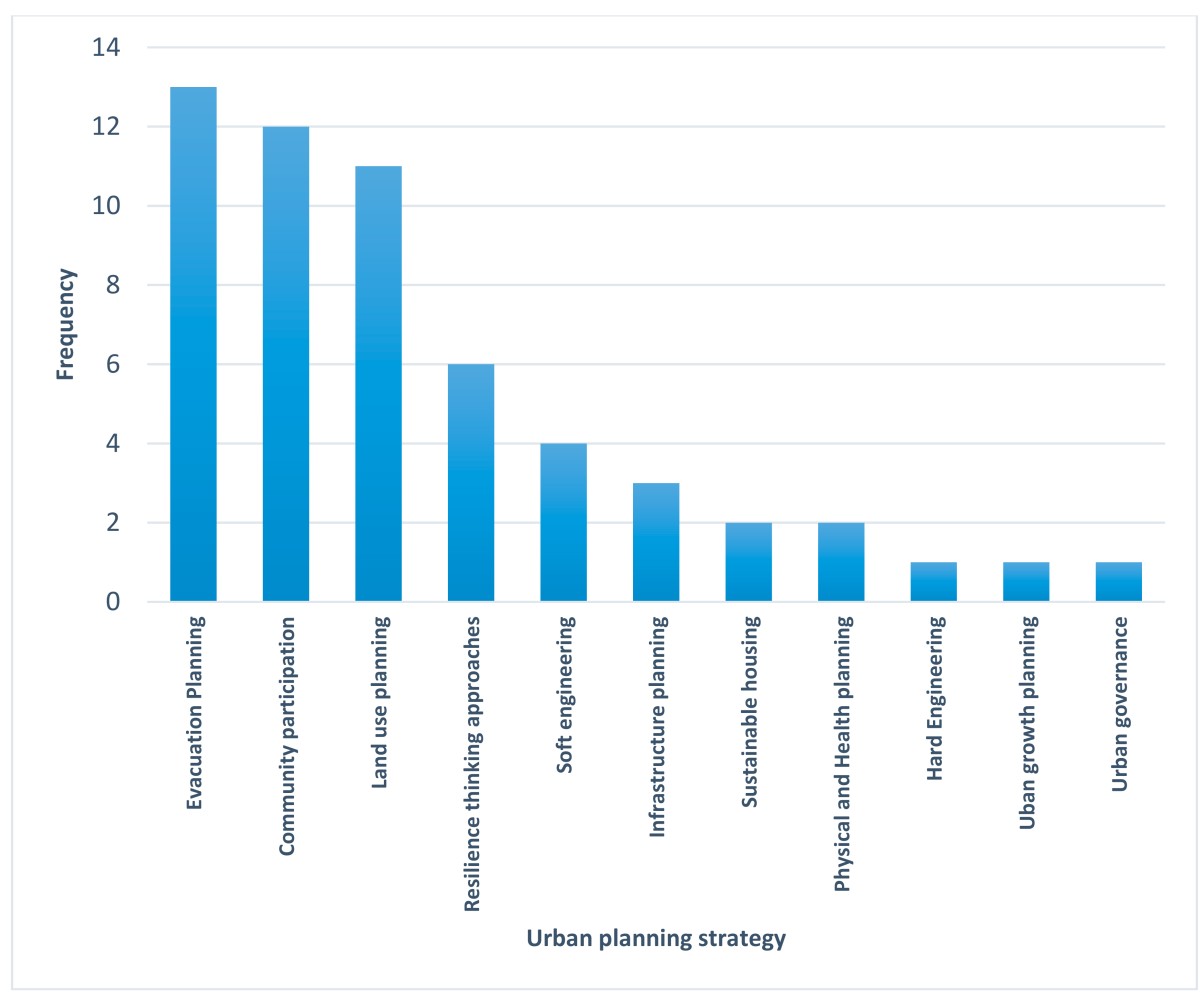

**Figure 7.** Existing strategies for urban planning in tsunami-prone areas.

Some studies have also suggested more innovative evacuation planning mechanisms, including personal life saving capsules [46], boat-based evacuation [47], and web-based interactive evacuation visualization tools [48]. These are suggested as ways to improve current evacuation planning strategies.

Most of the studies have focused on evacuation planning in the context of a single hazard event, rather than address any complications created by a multi-hazard scenario. This was also observed by Usha [49], in a study of coastal cities in India, who noted that flood evacuation mapping had been prepared without consideration of tsunamis and other coastal hazards.

Community participation was the second most discussed strategy (*n* = 12). Sustainable community engagement in the predisaster planning and post-disaster recovery processes were commonly cited themes. Many studies stressed the involvement of vulnerable communities. Spatial planning also featured prominently (*n* = 10). Among these studies, land use planning using digital technologies were the main focus. Nature-based planning (*n* = 3), structural planning (*n* = 3), and resilience thinking (*n* = 3) were among the other strategies discussed in the urban planning literature. These also tended to focus on single hazard events, rather than address multi-hazard scenarios.

Other social and economic aspects, including population growth planning (*n* = 1) and health-related planning (*n* = 1) have typically been ignored in the existing urban planning literature related to disaster risk reduction. However, the COVID-19 pandemic has again revealed the importance of health sector planning in these regions, where highly populated coastal cities were directly impacted.

As revealed by the reviewed sources, some of the urban planning approaches have become more prominent in recent years. These include urban resilience, population growth planning [50], and physical and social infrastructure planning [51,52]. These strategies tend to focus more on the demographical and social aspects in urban planning, for example, where the urban population growth must be controlled and managed effectively to achieve sustainability in these cities.

The following sections examine in more detail each of the main strategies (RQ2), as well as gaps and limitations that are revealed within the literature (RQ3).

### 3.2.1. Spatial Planning

Spatial planning is a main tool for urban planning that many governments use to influence the urban form and spatial development, including in tsunami-prone areas. The reviewed sources revealed land use planning (*n* = 4) as a major instrument for tackling disaster risk. Land use planning serves to demarcate zones or reserve land as buffers, which can make cities more resilient to the threats posed by a range of hazards [53]. Risk zone planning, which imposes restrictions for construction and development of infrastructure in specified zones, can be used to prepare coastal cities to cope with hazards such as tsunamis and coastal storms [54]. The literature also suggests that the inclusion of public open spaces and green spaces can be used to further empower the disaster resilience of cities while providing other ecosystem services for communities [55,56]. Similarly, Arif [57] suggested that the development of green open space, along with other zones such as commercial districts, residential areas, industrial areas, and transport networks, could influence the ecological balance as well as the economic balance of cities. Meutia [58] further discussed this concept with heritage site planning in a post-disaster context, where it could play an important role in the production of essential daily social functions, such as the provision of space for community consultation, conflict resolution, decision making, and disaster mitigation learning.

The reviewed sources show that governments can use land use planning strategies by empowering land use regulations through legislation and policy, and adopting planning instruments such as governmental statutes, regulations, rules, codes, and policies that can influence land use prior to a disaster event. For example, the relocation of residential developments in tsunami-prone zones or river floodplains, strongly decreases the vulnerability to a tsunami or river flooding [37]. In this sense, urban planning has strong potential for disaster risk reduction. Urban planning can set out land use regulations, including no-construction zones, and control development in hazard zones, and it can also define policies regarding evacuation, mandatory insurance, relocation of critical infrastructure, etc., which can notably reduce vulnerability to socio-natural disasters.

On a broad level, these planning instruments deal with the type, location, and extent of the land that is vulnerable to hazard events. Such vulnerability and risk assessments were frequently discussed in the reviewed sources ($n = 6$), especially with the application of digital technologies such as geographic information systems (GIS) and remote sensing. The development of GIS-based technologies and utilization of its tools for spatial planning were identified in several of the reviewed studies ($n = 4$). These technologies can be used to better understand hazards and to map vulnerability to reveal risk [49,59], which includes the physical and human factors such as topography, hydrography, land use, construction, settlements and communication networks in urban planning processes. Vulnerability assessments with digital technologies were also highlighted in the literature as an effective way to mitigate the impact of earthquakes and tsunamis [12,49,53]. Moreover, the utilization of satellite imagery in remote sensing and web-based risk assessment have provided new avenues for land use planning, which were highlighted in the studies as the next step of spatial planning for disaster risk reduction [59]. These technologies have been frequently used in tsunami and flood vulnerability mapping, while their use in connection with other hazards has been less evident.

### 3.2.2. Community and Key Stakeholder Participation

As with various types of spatial planning, consulting with the community and relevant stakeholders are an important part of land use planning to ensure transparency and incorporate a wide range of interests into the overall urban plan. This was highlighted in most of the studies, where communities and key stakeholder groups play significant roles in the urban planning process ($n = 12$). The concept of participation is seen as relevant as both a means and as an end.

Participation as a means aims at more effective implementation of urban planning programmes and projects through active citizen involvement in project implementation, usually by means of labour and/or financial or in-kind contributions [42,60]. Participatory risk zone planning is an example in Japan highlighted by Aoki [60], whereby the reconstruction process was empowered through local community participation in a more effective manner.

Participation as an end implies that citizens come up with ideas, take part in the decision-making process, assume responsibility, and finally arrive at self-management. Baudoin [61] revealed the importance of a community centric approach, where the grass-roots level and vulnerable communities paved the way for the design and application of early warning systems in an urban environment.

Several studies emphasised the inclusion of different layers of the community in the planning process, especially vulnerable and marginalized groups, as well as different age groups, genders, and ethnicities [54,61]. Castro [51] further discussed this with a specific approach to work with population with disabilities, noting the importance of incorporating them in all disaster risk and emergency management processes.

Institutional collaboration is also highlighted as a major concern for planning and implementation of urban planning strategies. For example, government should empower the institutions and facilitate them in the process [62]. Two studies highlighted the importance of a common platform, where experts and their institutions could present their plans, findings, and innovations in urban planning [63,64]. As highlighted by Djalante [63], effective participation and meaningful collaboration among all stakeholders, governments, NGOs, and communities is an important element of disaster risk reduction, where the increased recognition of the roles and responsibilities of local stakeholders in managing disaster risks is crucial. Similarly, the media is also recognised as a stakeholder that can, for example, be used to empower the early warning, awareness, and hazard education systems, which can each play an important role in urban planning for tsunami-prone regions [39].

### 3.2.3. Resilience Thinking Approaches

Numerous studies have suggested resilience thinking to support emergence of the resilient city concept, such as through the development of appropriate capacities of the urban system based on a combination of "absorbing disturbances and achieving a balance". An example of this is nature-based solutions ($n = 5$). The green city concept by Arif [57], the Waju community model by Ueda [65], and the build back greener approach by Mabon [66], all highlighted the importance of nature-based solutions in urban disaster resilience as a way to reduce disaster risk in tsunami-prone areas. As suggested by Arif [57], the urban development process should be conducted in a planned and integrated way, with special attention to spatial and environmental aspects to ensure an efficient urban management. The approach was further discussed with an ecological perspective by Ahern [67], with five strategies to build resilience capacity and transdisciplinary collaboration: biodiversity, urban ecological networks and connectivity, multifunctionality, redundancy and modularization, and adaptive design.

Community participation was also included in three studies as a key component in developing resilience. For example, Kennedy et al. [68] proposed a resilience thinking approach, as well as community and stakeholder involvement. They suggested the phrase "build back safer" instead of the "build back better", as they argued that the idea of "better" had multiple interpretations.

Ueda [66], in their study of the "Waju" community in Japan, stressed the importance of traditional knowledge, which must be combined with modern advanced technologies. They suggested that this traditional knowledge could be used to promote autonomy for disaster prevention and local supply of infrastructure and logistics for adaptation to hazardous environments.

Most of the reviewed sources stressed a "thinking ahead approach", integrating relief and development through long-term disaster risk reduction with sustainable urban development processes. Similarly, the knowledge required for approaching this urban sustainability and resilience can evolve rapidly in an adaptive planning and design context, as a complement to urbanization processes and projects in coastal regions.

### 3.2.4. Hard and Soft Engineering

Hard engineering management involves using artificial structures, whereas soft engineering management is a more sustainable and natural approach to manage coastal areas, such as to tackle coastal erosion [69]. The literature identified a range of potential countermeasures for disaster risk reduction in these broad categories: (1) hard engineering, containing seawall, groynes, and other hard engineered structures [70]; (2) soft engineering, including beach nourishment, mangrove afforestation [57,71,72], coral reef transplant [73], and coastal forest plantation [71]; (3) combined measures, such as beach nourishment seawall/groynes/breakwaters [74].

As per the reviewed sources, seawalls and manmade barriers such as groynes have been frequently used in coastal regions (*n* = 3). Nateghi [71] further explained that large seawalls have been shown to have been effective at reducing both mortality and damage rates in a hazard event, but smaller seawalls (around 5 m high) showed no effectiveness in reducing impact. However, there were a lack of studies on utilizing other concrete or wooden structures under hard engineering in these regions. However, as highlighted in some studies, soft engineering and/or combined structures are increasingly popular worldwide. For example, these can involve beach nourishment and biological restoration [73]. Among them, mangroves and coral reefs were prominent features in the reviewed sources (*n* = 5). Takagi [73] concluded that the impact of coastal floods could be substantially mitigated by planting a mangrove belt in front of a dyke. Soft engineering structures can also allow for sediment movement along the shoreline and often try to replicate natural processes. Some of the studies revealed the utilization of nature-based solutions with soft engineering structures as a proven sustainable solution for urban planning in tsunami-prone areas [73]. These authors contend that nature-based solutions have a prominent role in building resilience to future tsunamis, and can simultaneously act as a site for education and memorialisation. Further, coastal forests, coral reefs, and compacted dunes and wet lands in these urban areas could be incorporated as natural buffers for hazard mitigation, which could provide long-term sustainable solutions to face these extreme scenarios [54].

*3.3. Urban Planning in Tsunami-Prone Areas of Sri Lanka*

This section considers the findings in relation to RQ2 and RQ3 from the empirical investigations in Sri Lanka. Sri Lanka suffered the second highest losses in absolute terms from the 2004 Indian Ocean Tsunami. Coastal cities of the country were the most affected areas during the event including Ampara, Galle, Matara, Colombo, and Hambanthota [75]. The need for better urban planning to address disaster risk in the coastal cities of Sri Lanka was stressed in several reviews following the disaster. The section begins with an overall reflection of the urban planning strategies being adopted in Sri Lanka. The subsequent subsections discuss in more detail each of the main urban planning strategies, as well as gaps and limitations in the strategies which are identified through the expert interviews and focus group discussion.

Similar to the international literature, respondents pointed to evacuation planning and spatial planning as the major urban planning strategies in Sri Lanka to address disaster risk in coastal regions. For example, evacuation planning has been empowered with community awareness programmes, especially after the 2004 Indian Ocean Tsunami. This includes periodic drills in tsunami-prone areas since 2005. However, several disaster management professionals cautioned that such awareness programmes and drills had been badly disrupted or discontinued due to COVID-19.

The country has also implemented buffer zone regulations and guidelines as a spatial planning strategy for tsunami preparedness. However, as revealed by several of the civil engineering and disaster management professionals, under a new set of rules envisaged by the government, the buffer will be reduced to between 55 and 25 m in the southern districts and from 100 to 50 m in the northeast. Seven respondents also commented on the weak implementation of these regulations in coastal regions in Sri Lanka. For example, a civil engineer noted, "If there are regulations, planners have to adhere. But when it comes to guidelines, some can follow and some can skip" (C1). Further, an architecture professional stated that, "because of the political influence, they misinterpret and get away without following them".

The interviews also revealed that most of the planning strategies address a single hazard event, but do not adequately address multi-hazard scenarios. All twelve experts highlighted that law making and implementing authorities had tended to focus on single hazard events, with many attributing this to the different hazard profiles across the country, and many regions facing a dominant hazard that had become a focus for their preparedness

efforts. Nine of the respondents agreed that a multi-hazard approach is important in planning. Several used the example of COVID-19 to illustrate why a multi-hazard approach is needed, as the country had experienced flooding and coastal storms during the pandemic. Several noted that Sri Lankan legal frameworks did not adequately address pandemic preparedness and multi-hazard scenarios into planning.

Many of the respondents recognised a need to combine hard and soft engineering strategies in coastal regions as a way to reduce disaster risk associated with hazards such as tsunamis, floods, and coastal storms. However, several town planning and architecture-related professionals in particular stressed that, currently, there is an underutilisation of soft engineering and nature-based solutions in tsunami-prone areas. A disaster management professional explained (D2), "People look for short term solutions rather than the long-term benefits in planning". This was further explored in the focus group discussion, which revealed that land scarcity in coastal regions and the mindset of people and the government are the main reasons behind weak adoption of these approaches.

### 3.3.1. Spatial Planning in Sri Lanka

The importance of spatial planning in tsunami-prone areas has been recognised in Sri Lanka, and land use regulations have been introduced in such regions. Guidelines for housing development in coastal areas of Sri Lanka highlighted statutory requirements and best practice guides for settlement planning, housing design, and service provision for hazard preparedness in general. As observed in many international studies, digital technologies have also frequently been used in Sri Lanka for hazard and vulnerability mapping. The maps have been developed by the National Building and Research Organization and similar national level organizations. For tsunamis specifically, shoreline analysis, risk assessments, and vulnerability mapping exercises were also performed in coastal regions for disaster risk reduction after the Indian Ocean Tsunami. Despite such maps being developed, ten of the respondents highlighted that the maps are typically not used at the local authority level. The focus group discussion further highlighted that this was likely due to a combination of poor understanding and knowledge among local officials, but also insufficient effort to provide access to such maps for officials or the wider public. Further concerns were expressed about the resources available to update such maps, noting that many were developed over 15 years ago, often with the support of international agencies after the 2004 tsunami.

Another widely expressed concern was the weak implementation of building and planning regulations by developers, and a lack of monitoring by the local authorities. A town planning specialist (T2) summarised this as, "Planning and plan implementation are different" and "enforcers are abusing the power". Further, an architect and town planner (A2) suggested the absence of clear policy can give builders the basis to ignore the guidelines.

However, some also highlighted resource constraints as a major barrier to implementation. For example, several noted that land scarcity, cost, and a lack of expertise among local authorities as the barriers for proper land suitability analysis and spatial planning prior to construction. An experienced environmentalist and town planner (T1) who had worked in local authorities elaborated, "For smaller buildings, soil testing is a great cost. For smaller lands, the recommended shape of the building cannot be followed. Land scarcity and financial constraints are the barriers for proper implementation".

### 3.3.2. Community and Stakeholder Participation in Sri Lanka

In common with the wider literature, all respondents agreed that stakeholder participation is very important in urban planning in order to effectively address disaster risk. However, many doubted the effectiveness of current approaches in Sri Lanka. Public consultations were identified by many respondents as the prevailing platform for the public to comment on development proposals in the country, and this was recognised as a fundamental step in participatory risk zone planning. An architecture professional (A2) noted

that, "Public consultation is happening, but mostly through conventional interest group meetings". In order to increase the reach of such consultations, during the focus group it was suggested that digital platforms, including social media, could be better utilised in this process.

Poor access to risk information was also highlighted as a barrier to participation. Respondents tended to concur that the general public had no easy access to hazard, vulnerability, or risk information. Some respondents suggested that this might limit the ability for the public to make informed contributions to the planning process as they might not have access to the evidence that could be used to support their concerns.

Several respondents also highlighted the inadequate consideration of vulnerable communities in the planning process, for example, a failure to engage marginalised groups, people living in informal settlements, and people with disabilities. During the focus group, it was suggested that these people, as well as communities more widely, would benefit from greater engagement during the preplanning phases of development.

Some respondents warned that public consultation was currently hindered by political influence in the regions, although they did not feel comfortable elaborating on their concerns.

Greater involvement and accountability of related industries, fisheries, and tourism in the planning process was recommended by many experts. For example, one green building specialist and town planner (G1) suggested that, "Industries should be provided with and subject to enforcement against clearer regulations. This would force them to be more educated and prepared to self-analyze the impacts of their activities along the coastline".

Reflecting upon many of these concerns, during the focus group there was recognition of a need to re-engineer the stakeholder engagement processes, and provide better platforms to facilitate such engagement.

### 3.3.3. Resilience Thinking Approaches in Sri Lanka

Many cities in Sri Lanka have joined the UN Making Cities Resilient campaign, while similar to many other countries that have experienced major disasters in their recent history, Sri Lanka has also implemented a "build back better" approach in development, especially after the 2004 tsunami. More recently, this has been refined to be a "build back safer" approach to promote the long-term protection of the community. Although related terms were frequently used during the interviews and focus group, several respondents were sceptical and indicated that only a few major projects, such as the Galle City Development Project, properly adhered to these principles. They noted that many local authority level projects, especially those without international donor support, did not adopt "build back better" or other resilience-based approaches at the city or project levels.

The "living with floods" concept is visible in a few rural areas, where native solutions and indigenous knowledge have been integrated within survival planning. Despite some isolated examples, a majority of respondents felt more effort should be made to draw upon lessons from the past, especially drawing upon local knowledge. Several noted that current planning approaches are top-down, and fail to adequately draw upon local experiences when developing resilience. The importance of combining scientific and traditional knowledge was also highlighted by some. For example, an Architect (H1) suggested, "Cases can be found in rural coastal cities where tidal patterns, flood information, and mitigation measures have been adopted from local knowledge. Boat evacuation during floods also shows the validity of traditional methods even in the present context". Other resilience examples cited by the experts included the ancient cascading system, ecological restoration. and community resilience with social capital that could be effective in an urban planning context.

In order to promote greater resilience thinking in urban planning, several respondents stressed the importance of leadership. This was especially so for major city development projects and national level planning, which invariably require the support of senior government officials. A disaster management specialist (D1) proposed that, "Policy makers, politicians, and planners should get together and do proper planning after proper analysis. Good leaders will enable good resilience thinking approaches".

### 3.3.4. Hard and Soft Engineering Approaches in Sri Lanka

Sri Lanka has adopted a range of hard and soft engineering strategies to protect its coastal regions. Groynes and sea walls are among the major hard engineering approaches that have been used, along with soft strategies such as beach nourishments and mangrove plantations. However, most experts noted the absence of many large hard engineered structures.

One structural engineer (C2) explained, "Sri Lanka doesn't need many coastal protection structures—there are cost constraints and tsunami is such a low frequency event. It would be difficult to justify. Instead, hazard resilient building construction should be given priority".

The importance of having an integrated building code was stressed by all the structural engineering respondents. However, one (C3) highlighted some limitations in the current approach, including a lack of consistency and legal obligations, "Several guidelines and frameworks from different authorities which are not legally binding will not serve the purpose".

Recently, nature-based solutions have been introduced as a more sustainable solution by the respective agencies. Some examples of ecological restoration programmes in coastal regions were highlighted by respondents as evidence of the emerging shift towards nature-based solutions, and of further opportunities to make better use of existing natural resources to support safer urban planning in tsunami-prone areas. These approaches appear to be popular with authorities as they are often viewed as cost effective when compared with hard engineering. Despite this, one respondent cautioned that there are often barriers, and gave the example of a protection of plantation scheme that was at risk due to land ownership issues and a lack of understanding among communities. Competing stakeholder interests were also seen as a barrier. For example, five respondents stressed the need for more stringent legal frameworks to protect natural assets that were providing protection, including beaches and coral reefs. One expert suggested that, "There must be dedicated personnel at local authorities with the authority to monitor the preservation of beach and nature-based solutions, and report to the authorities to take actions against those who violate the laws".

## 4. Discussion

In order to bring the literature and empirical findings together and address RQ2, Figure 8 illustrates some of the key urban planning strategies that are currently used in tsunami-prone areas before a hazard event occurs, in the immediate response to a hazard event, and during recovery and reconstruction following a disaster. The specific urban planning approaches are broadly grouped into six high-level strategies: community participation, spatial planning, soft and hard engineering, evacuation planning, and resilience thinking. Some of these strategies are specific to one phase, while other strategies cross multiple phases, although the specific urban planning approaches within each strategy tend to differ across them. Within each strategy, examples of the types of approaches are presented.

RQ3 sought to identify gaps or limitations in these existing approaches. The following discussion attempts to draw out some common themes to emerge.

The importance of community participation in planning processes is widely recognized in the literature across all three phases, i.e., before, responding, and recovering from disaster. Despite also recognizing its importance, existing approaches in Sri Lanka have often failed to reach members of the community, especially marginalized and vulnerable groups. The methods of engagement are limited in nature and there has been little attempt to increase the reach, such as through new technology or social media. Most of the reviewed sources have also highlighted the importance of stakeholder engagement in a broader sense and to provide a realistic picture in grassroots level disaster risk reduction. The integration of different stakeholders' perspectives is a particular concern in the planning process, and many studies, including Sri Lanka, have highlighted the potential for conflict due to different interests.

Similarly, a lot of disaster risk information is gathered in Sri Lanka, but is difficult to access, and often not used to inform development planning. It is also evident that although there was some investment in technology and the development of risk information following the 2004 tsunami, there has been only limited use of GIS and remote sensing, or resources available to adopt more recent developments such as web-based planning and better use of real-time information. This reaffirms the challenges such countries face in developing sustainable approaches to urban planning. Although they can benefit from substantial external assistance following a major disaster, the country is often left without the capacity to sustain these efforts and, over time, the quality and timeliness of such information diminishes.

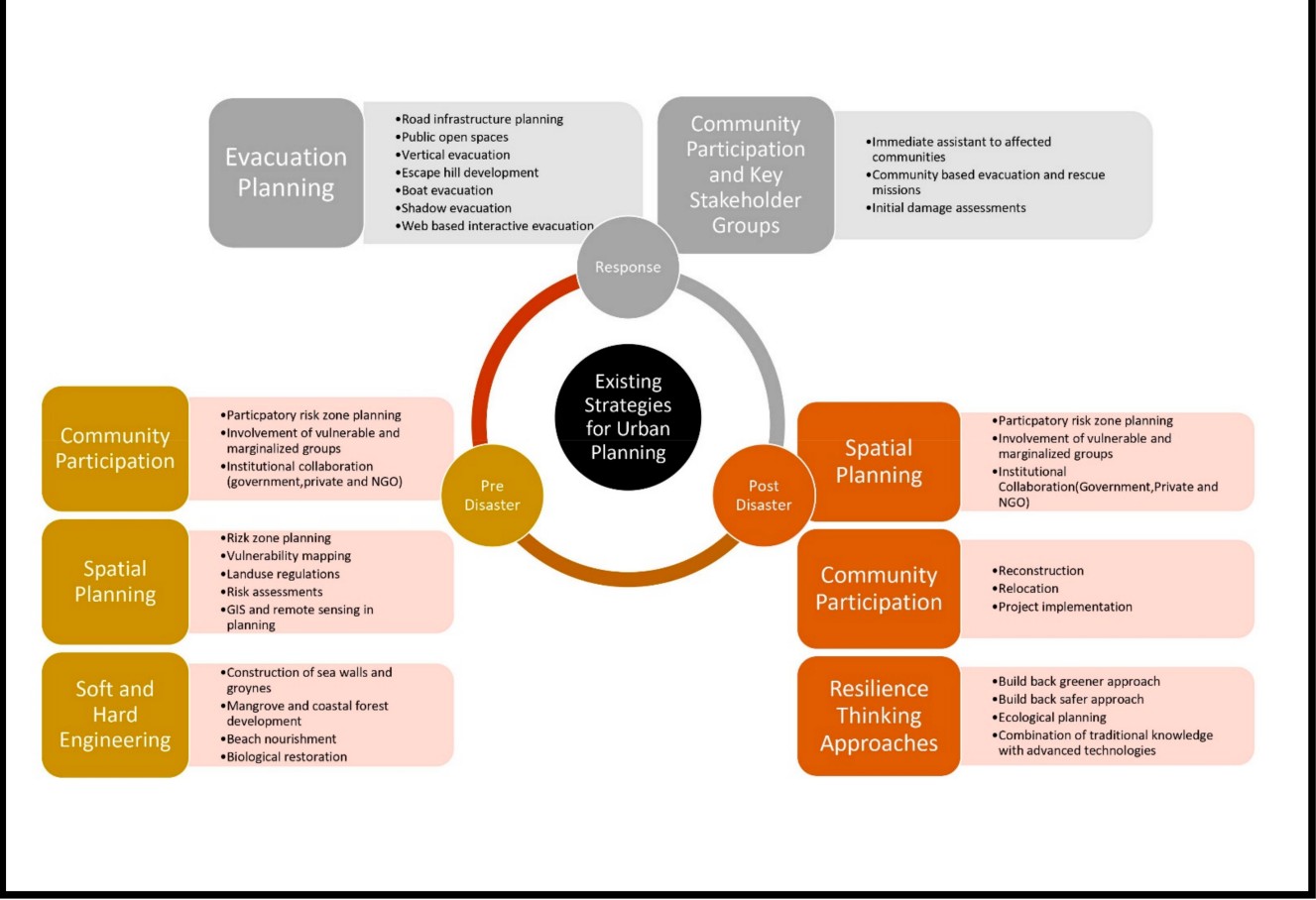

**Figure 8.** Existing strategies for urban planning in tsunami-prone areas.

Among the literature, the lack of health- and population-related planning strategies in the predisaster phase is notable, for example, locational and population growth planning. This is also the case in Sri Lanka. For example, the role of population planning aspects in highly populated coastal cities including Colombo, Kalutara and Galle, does not appear to have been actively considered in relation to disaster risk.

A combination of soft and hard engineering approaches can be effective in reducing disaster risk in tsunami-prone areas. The potential for hard engineering structures is well established, although in many situations, the cost is prohibitive, especially when facing very infrequent hazard events, which makes it difficult to justify. Sri Lanka appears to have embraced some soft engineering, including nature-based solutions that are increasingly being promoted for their potential benefits in biodiversity protection, safety, and cost effectiveness. However, there are concerns about the extent to which such interventions are being maintained and the capacity of local officials to protect them.

Even though resilience thinking approaches were discussed in the literature, most of the existing approaches were limited to ecological resilience or build back better approaches after a disaster. Social and physical aspects were not included in most of the studies [73]. To facilitate a holistic interpretation of urban planning in these coastal regions, it may be necessary to employ different aspects of resilience in the existing resilience approaches and methods as well. For instance, assessing ecosystem services of a neighborhood's green infrastructure may reveal the performance of a particular urban area in relation to biodiversity protection, public open space utilization, and local level engagement. This supports the view of Ueda [65] who suggested that it would be important to open a space for DRR practitioners and scholars to pause for thought, to reconsider, and to reformulate. This might include providing opportunities for knowledge sharing and coupling of traditional and scientific knowledge in resilience thinking for better results.

The need for a more multi-hazard approach is widely recognized in the Sendai Framework for Disaster Risk Reduction. However, a commonality among previous studies from the literature and current practices in Sri Lanka is the primary focus on single hazard events, or at best, some consideration of cascading hazards, where one hazard, for example, an earthquake, triggers another hazard, a tsunami. The COVID-19 pandemic has again brought to the forefront the importance and challenges associated with multi-hazard threats. These can be especially challenging where they are concurrent but independently occurring hazards that interact. The nature of COVID-19, which has disrupted society for several years, has clearly exposed the potential for this type of situation. Many countries around the world have experienced other hazard events while tackling the pandemic threat. As highlighted in Sri Lanka, the COVID-19 response resulted in a shift of priorities, alterations in work practices and locations, the imposition of physical distancing, self-isolation and quarantine measures, as well as temporary lockdowns of entire communities. It will be important to understand how such changes might impact wider hazard vulnerability, and to what extent urban planning can reduce or exacerbate disaster risk in such a situation.

This also relates to the broader issue of systemic risk, which is highlighted in the 2022 Global Assessment Report [76]. The concept of systemic risk is based on the notion that the risk of an adverse outcome of a policy, action, or hazard event can depend on how the elements of the affected systems interact with each other. This can either aggravate or reduce the overall effect of the constituent parts. Interactions occur through positive or negative feedback processes. Systemic risk creates the chance of system malfunction or even collapse [77]. Urban planning has the potential to greatly influence risk-sensitive urban development in a manner that can transform the way cities are built to face the uncertainties that arise from climate-induced disaster risks, but also address the complexity of multi-hazard threats. However, there appears to be a dearth of previous studies that attempt to tackle such complexity, and there is little evidence in Sri Lanka that such approaches are permeating into policy and practice.

## 5. Conclusions

This systematic review of previous studies on urban planning in tsunami-prone areas, along with a more detailed examination of current urban planning approaches in Sri Lanka, provide some useful insights into the range of strategies available to reduce disaster risk, but also some of the many challenges associated with their implementation in policy and practice.

Tsunamis, as an infrequent, but potentially devastating hazard threat to many coastal cities, have prompted researchers and related policymakers to work on urban planning strategies that can reduce disaster risk in tsunami-prone areas. It is also evident from the literature and results to emerge from the study in Sri Lanka that the 2004 Indian Ocean tsunami, as well as other major tsunami events in recent years, have prompted increased investment and attention on urban planning efforts, as well as numerous studies to investigate specific approaches and their effectiveness. However, the lack of common methodological approaches across studies, as well as the lack of detail in the presentation of research methods, makes direct comparisons across studies more difficult. It also does not lend itself to research synthesis, which could otherwise be used to inform improvements to urban planning in different contexts.

The results also reveal there are many urban planning challenges to overcome in order to better protect tsunami-prone areas. The model in Figure 8 provides a useful way of illustrating the breadth of urban planning strategies and approaches that can be adopted to help tackle disaster risk in tsunami-prone areas. The model is also an initial attempt to provide a basis for further detailed empirical studies of urban planning in tsunami-prone areas, including much needed comparative studies. However, the authors recognize there is much work to be done in further exploring the model through more detailed empirical studies in different contexts and to ensure that current practices are adequately reflected.

This review is limited by its reliance on other scholars self-reporting results in their own studies and opinions from the different professionals in Sri Lanka. Furthermore, there are language inconsistencies across the literature in terms of what scholars identify as soft engineering, nature-based, and hard engineering. Therefore, the analysis of the literature is based on our interpretation of what is at times unclear work of others. Still, this review points to new research that would benefit the field of urban planning in tsunami-prone areas. Primarily, future research will need to establish guiding planning principles, build goal-oriented asassessment frameworks under these principles, and test the frameworks with empirical assessment studies. This should be pursued in parallel with the development of a common framework to unify urban planning literature. A common set of urban planning strategies can emerge for multi-hazard scenarios and to tackle systemic risk.

**Author Contributions:** Conceptualization, U.T.G.P., R.H. and D.A.; methodology, R.H. and D.A.; software U.T.G.P.; validation, C.D.Z. and A.A.S.E.A.; formal analysis, U.T.G.P.; investigation, U.T.G.P.; resources, U.T.G.P., C.D.Z. and A.A.S.E.A.; data curation, U.T.G.P. and C.D.Z.; writing—original draft preparation, U.T.G.P.; writing—review and editing, R.H.; visualization, U.T.G.P.; supervision, R.H., D.A. and R.D.; project administration, R.H., D.A. and R.D.; funding acquisition, R.H. and D.A. All authors have read and agreed to the published version of the manuscript.

**Funding:** The underpinning research was co-funded by UK Research and Innovation (UKRI) through the UK Government's Global Challenges Research Fund (GCRF) and the UK Newton Fund [grant number EP/V026038/1]. The underpinning research was co-funded by the UK Parliamentary Under Secretary of State for Business, Energy and Industrial Strategy (BEIS) through the Newton Prize and UK Newton Fund [grant number AM002715]. UKRI and BEIS accept no liability, financial or otherwise, for expenditure or liability arising from the research funded by the grant, except as set out in the Terms and Conditions or otherwise agreed in writing.

**Institutional Review Board Statement:** The research was approved by the ethics committee of the lead University that coordinated the research. Institution: University of Huddersfield School of Applied Sciences Research Integrity and Ethics Committee Code: SAS-SREIC 25.01.21-9.

**Informed Consent Statement:** Informed consent to participate in the study was obtained from participants in interviews and the focus group. This included information on how data would be protected, confidentiality and privacy, as well consent to publish results in an anonymized form. No personally identifiable information is included within this publication.

**Data Availability Statement:** The data that support the findings of this study are available on request from the corresponding author, U.T.G.P. The data are not publicly available due to their containing information that could compromise the privacy of research participants.

**Acknowledgments:** The authors gratefully acknowledge the contributions of experts in interviews and the focus group discussion.

**Conflicts of Interest:** The authors declare no conflict of interest.

## Appendix A

**Table A1.** Reviewed sources of the systematic review.

| ID No. | Paper | Title of the Journal and Published Source | Author | Year |
|---|---|---|---|---|
| 1 | Quantifying urban physical growth types in Banda Aceh City after the 2004 Indian Ocean Tsunami | E3S Web of Conferences | Amri, S.R. Giyarsih | 2020 |
| 2 | Green city Banda Aceh: City planning approach and environmental aspects | IOP Conference Series: Earth and Environmental Science | A.A. Arif | 2017 |
| 3 | Escape hill development as a strategy to improve urban safety after earthquake and tsunami Aceh 2004 based on regional planning and geotechnical aspect | *Journal of Physics*: Conference Series | M. Munirwansyah, H. Munirwan, M. Irsyam, R.P. Munirwan | 2020 |
| 4 | Resident's satisfaction to relocated Houses after 2004 Indian Ocean Tsunami, Thailand | *Procedia Engineering*, Elsevier | Titaya Sararita, Kondo Tamiyob, and Elizabeth Maly | 2017 |
| 5 | Integration of disaster management strategies with planning and designing public open spaces | *Procedia Engineering*, Elsevier | R.R.J.C. Jayakody, D. Amarathunga, R. Haigh | 2017 |
| 6 | Assessment of road traffic performance of the Tsunami evacuation road in Padang Municipality area based on the traffic volume simulation approach | E3S Web of Conferences | Y. Yossyafra, N. Fitri, R.P. Sidhi, Y. Yosritzal, D.I. Mazni | 2020 |
| 7 | Opportunities and Risks of the "New Urban Governance" in India: To What Extent Can It Help Addressing Pressing Environmental Problems? | *Journal of Environment & Development*, SAGE | Jeroen van der Heijden | 2016 |
| 8 | A Study on transformation of living environment and domestic Spatial Arrangements: Focused on a western coastal housing settlement of Sri Lanka after Sumatra Earthquake and Tsunami 2004 | *Journal of Asian Architecture and Building Engineering*, Architectural Institute of Japan | Woharika Kaumudi Weerasinghe and Tsutomu Shigemura | 2008 |

**Table A1.** *Cont.*

| ID No. | Paper | Title of the Journal and Published Source | Author | Year |
|---|---|---|---|---|
| 9 | Urban landscape sustainability and resilience: The promise and challenges of integrating ecology with urban planning and design **** | *Landscape Ecology*, Springer | Jack Ahern | 2013 |
| 10 | The Sendai Framework for Disaster Risk Reduction: Renewing the global commitment to people's resilience, health, and well-being | *International Journal of Disaster Risk Science*, Springer | Amina Aitsi-Selmi, Shinichi Egawa, Hiroyuki Sasaki, Chadia Wannous, Virginia Murray | 2015 |
| 11 | "Waju" and its evolution with urban technology-Japanese sustainable community for disaster resilience | IOP Conference Series: Earth and Environmental Science | R. Ueda | 2019 |
| 12 | Enhancing post-disaster resilience by 'building back greener': Evaluating the contribution of nature-based solutions to recovery planning in Futaba County, Fukushima Prefecture, Japan | *Landscape and Urban Planning* | L. Mabon | 2019 |
| 13 | From top-down to "community-centric" approaches to early warning systems: Exploring pathways to improve disaster risk reduction through community participation | *International Journal of Disaster Risk Science*, Springer | Marie-Ange Baudoin, Sarah Henly-Shepard, Nishara Fernando, Asha Sitati Zinta Zommers | 2016 |
| 14 | Disaster risk perception in urban contexts and for people with disabilities: case study on the city of Iquique (Chile) | *Natural Hazards*, Springer | Carmen-Paz Castro, Juan-Pablo Sarmiento, Rosita Edwards, Gabriela Hoberman, Katherine Wyndham | 2016 |
| 15 | Bangkok to Sendai and Beyond: Implications for disaster risk reduction in Asia | *International Journal of Disaster Risk Science*, Springer | Ranit Chatterjee, Koichi Shiwaku, Rajarshi Das Gupta, Genta Nakano, Rajib Shaw | 2015 |
| 16 | The Indian Ocean Tsunami: Economic impact, disaster management, and lessons | Asian Economic Papers-The Earth Institute at Columbia University and the Massachusetts Institute of Technology | Prema-chandra Athukorala | 2006 |
| 17 | The COVID-19 pandemic: Impacts on cities and major lessons for urban planning, design, and management | *Science of the Total Environment*, Elsevier | Ayyoob Sharifi, Amir Reza, Khavarian-Garmsir | 2020 |
| 18 | Flood risk and adaptation in Indian coastal cities: Recent scenarios | *Applied Water Science*, Springer | Ravinder Dhiman, Renjith VishnuRadhan, Eldho, Arun Inamdar | 2018 |
| 19 | Community resilience and urban planning in tsunami-prone settlements in Chile | *Disasters*, Overseas Development Institute | Marie Geraldine Herrmann-Lunecke | 2019 |

**Table A1.** *Cont.*

| ID No. | Paper | Title of the Journal and Published Source | Author | Year |
|--------|-------|-------------------------------------------|--------|------|
| 20 | Adaptive governance and managing resilience to natural hazards | *International Journal of Disaster Risk Science*, Springer | Riyanti Djalante, Cameron Holley, and Frank Thomalla | 2019 |
| 21 | Multi-criteria location planning for public facilities in tsunami-prone coastal areas | *OR Spectrum*, Springer | Karl F. Doerner, Walter J. Gutjahr, Pamela C. Nolz | 2008 |
| 22 | Validating a tsunami vulnerability assessment model (the PTVA model) using field data from the 2004 Indian Ocean Tsunami | *Natural Hazards*, Springer | Dale Dominey-Howes and Maria Papathoma | 2006 |
| 23 | From multi-risk evaluation to resilience planning: The case of central Chilean coastal cities | *Water* (Switzerland) | P. Barría, M.L. Cruzat, R. Cienfuegos, J. Gironás, C. Escauriaza, C. Bonilla, R. Moris, C. Ledezma, M. Guerra, R. Rodríguez, A. Torres | 2019 |
| 24 | Evaluation of the reconstruction plans for tsunami victims in Malaysia | *Journal of Asian Architecture and Building Engineering*, Architectural Institute of Japan | F.S. Ling | 2006 |
| 25 | The role of built environment's physical urban form in supporting rapid tsunami evacuations: Using computer-based models and real-world data as examination tools | *Frontiers in Built Environment*, Frontiers Editorial Office | Foong Sau Ling, Yoshimitsu Shiozaki, and Yumiko Horita | 2018 |
| 26 | Improved coastal zone planning and management | Integrated coastal zone planning in Asian tsunami-affected countries | Robert Kay | 2006 |
| 27 | Personal sky equipment for inhabitants of coastal cities: Envisioning an evacuation system to reduce disaster's impact during the climate change era | IOP Conference Series: Materials Science and Engineering | K. Januszkiewicz | 2019 |
| 28 | Urban resources selection and allocation for emergency shelters: In a multi-hazard environment | *International Journal of Environmental Research and Public Health*, Molecular Diversity Preservation Internationa (MDPI) | Wei Chen, Guofang Zhai, Chongqiang Ren, Yijun Shi, and Jianxin Zhang | 2018 |
| 29 | Challenges of post-tsunami reconstruction in Sri Lanka: Health care aid and the Health Alliance | CRISIS—FOR DEBATE, *The Medical Journal of Australia* (MJA) | Paul A Komesaroff and Suresh Sundram | 2006 |
| 30 | Assessing the impact of the Indian Ocean Tsunami on households: A modified domestic assets index approach | *Disasters*, Overseas Development Institute | Sudha Arlikatti, Walter Gillis Peacock, Carla S. Prater, Himanshu Grover, and Arul S. Gnana Sekar | 2010 |

**Table A1.** *Cont.*

| ID No. | Paper | Title of the Journal and Published Source | Author | Year |
|---|---|---|---|---|
| 31 | Sustainable downtown development for the tsunami-prepared urban revitalization of regional coastal cities | *Sustainability* (Switzerland) | T. Ito, T. Setoguchi, T. Miyauchi, A. Ishii, N. Watanabe | 2019 |
| 32 | Integrated approach for coastal hazards and risks in Sri Lanka | *Natural Hazards and Earth System Sciences*, Copernicus Publications on behalf of the European Geosciences Union | M. Garcin, J.F. Desprats, M. Fontaine, R. Pedreros, N. Attanayake, S. Fernando, C.H.E.R. Siriwardana, U. De Silva, and B. Poisson | 2008 |
| 33 | Tsunami hazard assessment of Chabahar Bay related to megathrust seismogenic potential of the Makran subduction zone | *Natural Hazards*, Springer | A.R. Payande, M.H. Niksokhan, H. Naserian | 2014 |
| 34 | Risk and vulnerability assessment to tsunami and coastal hazards in Indonesia: Conceptual framework and indicator development | Conference Paper, Research Gate | J. Post, K. Zosseder, G. Strunz, J. Birkmann, N. Gebert, N. Setiadi, H.Z. Anwar, H. Harjono, M. Nur, T. Siagian | 2014 |
| 35 | An integrated social response to disasters: The case of the Indian Ocean Tsunami in Sri Lanka | *Disaster Prevention and Management*, Emerald Group Publishing Limited | Siri Hettige and Richard Haigh | 2016 |
| 36 | Assessing people's early warning response capability to inform urban planning interventions to reduce vulnerability to tsunamis | *Institute of Geodäsey and Geoinformation*, Bonn University | N.J. Setiadi | 2014 |
| 37 | Tsunami vulnerability assessment in urban areas using numerical model and GIS | *Natural Hazards*, Springer | Tune Usha, M.V. Ramana Murthy, N.T. Reddy, Pravakar Mishra | 2011 |
| 38 | Measuring tsunami planning capacity on US Pacific coast | *Natural Hazards Review*, Infrastructure Resilience Division | Z. Tang, M.K. Lindell, C.S. Prater, and S.D. Brody, 2008. | 2008 |
| 39 | Sequencing and combining participation in urban planning: The case of tsunami-ravaged Onagawa Town, Japan | *Cities*, Elsevier | N. Aoki | 2018 |
| 40 | Urban planning and tsunami impact mitigation in Chile after February 27, 2010 | *Natural Hazards*, Springer | M.G.H. Lunecke | 2015 |
| 41 | The meaning of 'build back better': Evidence from post-tsunami Aceh and Sri Lanka | *Journal of Contingencies and Crisis Management* | J. Kennedy, J. Ashmore, E. Babister, and I. Kelman | 2008 |
| 42 | Environmental implications for disaster preparedness: Lessons Learnt from the Indian Ocean Tsunami | *Journal of Environment Management*, Elsevier | H. Srinivas | 2008 |

**Table A1.** *Cont.*

| ID No. | Paper | Title of the Journal and Published Source | Author | Year |
|---|---|---|---|---|
| 43 | A systematic study of disaster risk in brunei darussalam and options for vulnerability-based disaster risk reduction | *International Journal of Disaster Risk Science*, Springer | Anthony Banyouko Ndah, John Onu Odihi | 2017 |
| 44 | Disaster waste clean-up system performance subject to time-dependent disaster waste accumulation | *Natural Hazards*, Springer | Cheng Cheng, Lihai Zhang, Russell George Thompson, Greg Walkerden | 2017 |
| 45 | Identification and classification of urban micro-vulnerabilities in tsunami evacuation routes for the city of Iquique, Chile | *Natural Hazards and Earth System Sciences* | G. Álvarez, M. Quiroz, J. León, R. Cienfuegos | 2018 |
| 46 | Heritage planning and rethinking the meaning and values of designating heritage sites in a post-disaster context: The case of Aceh, Indonesia | IOP Conference Series: Earth and Environmental Science | Z.D. Meutia, R. Akbar, D. Zulkaidi | 2018 |
| 47 | Assessing tsunami vulnerability areas using satellite imagery and weighted cell-based analysis | *International Journal of GEOMATE* | Guntur, A.B. Sambah, F. Miura, Fuad, D.M. Arisandi | 2017 |
| 48 | Mangrove forest against dyke-break-induced tsunami on rapidly subsiding coasts | *Natural Hazards and Earth System Sciences* | H. Takagi, T. Mikami, D. Fujii, M. Esteban, S. Kurobe | 2016 |
| 49 | A household-level flood evacuation decision model in Quezon City, Philippines | *Natural Hazards*, Springer | Ma. Bernadeth, Lim, Hector R., Lim Jr, Mongkut Piantanakulchai, Francis Aldrine Uy | 2016 |
| 50 | Visual exploration of tsunami evacuation planning | *Journal of the Visualization Society of Japan* | Cui Xie, Guangxiao Ma, Qiong Li, Jinjin Xun, Junyu Dong | 2015 |
| 51 | The 2011 Tohoku Tsunami: Implications for natural disaster management in Japan | *Revista INVI* | Y.C.C. Gatica, M.B. Benítez | 2015 |
| 52 | Statistical analysis of the effectiveness of seawalls and coastal forests in mitigating tsunami impacts in Iwate and Miyagi Prefectures | *PLOS ONE* | Roshanak NateghiJeremy D. Bricker, Seth D. Guikema, Akane Bessho | 2016 |
| 53 | Natural hazards, vulnerability and structural resilience: Tsunamis and industrial tanks | *Geomatics, Natural Hazards and Risk*, Informa UK Limited | Ahmed Mebarki, Sandra Jerez, Gaetan Prodhomme, and Mathieu Reimeringer | 2016 |
| 54 | Influence of road network and population demand assumptions in evacuation modeling for distant tsunamis | *Natural Hazards*, Springer | Kevin D. Henry, Nathan J. Wood, Tim G. Frazier | 2016 |

**Table A1.** *Cont.*

| ID No. | Paper | Title of the Journal and Published Source | Author | Year |
|--------|-------|-------------------------------------------|--------|------|
| 55 | Disaster risk reduction including climate change adaptation over South Asia: Challenges and ways forward | *International Journal of Disaster Risk Science*, Springer | Rajesh K. Mall, Ravindra K. Srivastava, Tirthankar Banerjee, Om Prakash Mishra, Diva Bhatt, Geetika Sonkar | 2019 |
| 56 | Tsunami risk reduction for densely populated Southeast Asian cities: Analysis of vehicular and pedestrian evacuation for the city of Padang, Indonesia, and assessment of interventions | *Natural Hazards* | M. Di Mauro, K. Megawati, V. Cedillos, B. Tucker | 2013 |

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
