# Peer review of "A Study of Urban Planning in Tsunami-Prone Areas of Sri Lanka"

_2673-8945, doi:10.3390/architecture2030031_

Round 1
Reviewer 1 Report
Overall, this article is well-structured and provides a lot of material. However, consider the following suggestions:
1. Try to consolidate and simplify the content of the article.
2. Be sure to strengthen the layout of the article, the resolution of figures and tables.
Author Response
|
Reviewer 1
Thank you for the constructive feedback, which we accept and have revised the article as detailed below, and in the manuscript that indicates all revisions as ‘track changes’. |
||
|
ID.No |
Comment |
Response |
|
01. |
Try to consolidate and simplify the content of the article. |
Revise the result section according to the three research questions. Categorise the result section into three main sections; Results on systematic review and Interview results from Sri Lanka 3.1 Characteristics of the literature 3.2 Existing strategies for urban planning in tsunami-prone areas Existing strategies for urban planning in tsunami-prone areas
3.3 Urban planning in tsunami prone areas of Sri Lanka
Divide the discussion and conclusion into two section (Section 4 and 5). |
|
02. |
Be sure to strengthen the layout of the article, the resolution of figures and tables. |
Enhance the layout and resolution of figures (Figure 1-7) and tables (Table 1) |

Reviewer 2 Report
The work deals a topic of great interest in a comprehensive way and can make a contribution to the state of the art and related issues. The critical analysis carried out of the contributions taken as a reference offers a clear and linear update.
In the introduction, the objectives are clear to the reader. Today it is important to have a framework that helps to understand and evaluate how urban planning strategies can support cities to face extreme events that can occur simultaneously, whether these are a pandemic, a tsunami, an earthquake or other.
The methodology can be of help to the subjects to whom it is addressed and is coherent and well explained in each step.
The discussion is well supported by the results they have obtained and the objectives have been achieved. Each theme treated in the method is taken up and translated into material useful for reaching conclusions, the steps are linear.
The conclusions are indicative with respect to the objectives, I appreciate the fact that the author has highlighted the limitations of his study, and the valid justification is considered, and the interesting contribution with respect to the developments outlined for the next research steps. It would be interesting a graph that aims to graph the next steps foreseen. Of course it is only a suggestion. However, the conclusions remain clear even without this step.
In general, in various points of the text there are line errors, which is why I suggest a revision in this sense.
In my opinion, the work can be published after the minor revisions that refer to the aforementioned line errors.
Author Response
|
Reviewer 2
Thank you for the constructive feedback, which we accept and have revised the article as detailed below, and in the manuscript that indicates all revisions as ‘track changes’. |
||
|
ID.No |
Comment |
Response |
|
01. |
In general, in various points of the text, there are line errors, which is why I suggest a revision in this sense. |
Corrected the line errors and typo errors in the document |

Reviewer 3 Report
Dear authors,
The work presented is of great interest, but some aspects are still not sufficiently detailed and require further attention.
The work complies with the requirements of the special issue, but the originality of the work is not clear, there are some typos (for example, in line 13 "Ef-fective") and the whole part on COVID- 19 is very weak and slightly out of context with respect to the general work.
The whole paragraph 3 is very complex to read, and I suggest restructuring the entire paragraph in a more readable way and with a more accurate iconographic apparatus.
I would also suggest explaining better what the original contribution of the interviews is with respect to the specific literature review conducted with the PRISMA Analysis.
In general, a clearer description of the results, with direct references to the studies analyzed, would be desirable. At the moment, in fact, the entire third paragraph is extremely chaotic and unstructured.
Also I would divide the discussions by the conclusions.
I hope my comments are useful to the authors.
Author Response
|
Reviewer 3
Thank you for the constructive feedback, which we accept and have revised the article as detailed below, and in the manuscript that indicates all revisions as ‘track changes’. |
||
|
ID.No |
Comment |
Response |
|
01. |
There are some typos (for example, in line 13 "Ef-fective") and |
Corrected the typo errors in abstract and done another proof read |
|
02. |
The whole part on COVID- 19 is very weak and slightly out of context with respect to the general work. |
The paragraph on COVID-19 was removed from the introduction section and strengthen the section on urban planning and disaster risk (1.3) |
|
03. |
The whole paragraph 3 is very complex to read, and I suggest restructuring the entire paragraph in a more readable way and with a more accurate iconographic apparatus. |
Section 1.3 was revised after the removal of the paragraph on COVID – 19.
Whole result section (3) was restructured according to the three modified research questions. Respectively 3.1 Characteristics of the literature 3.2 Existing strategies for urban planning in tsunami-prone areas Existing strategies for urban planning in tsunami-prone areas
3.3 Urban planning in tsunami prone areas of Sri Lanka
|
|
04. |
I would also suggest explaining better what the original contribution of the interviews is with respect to the specific literature review conducted with the PRISMA Analysis. |
Included a new paragraph on Materials and methods section (2) to emphasise the original contribution of Interviews (Page No.06)
Included a new table (Table 1) on Interview respondent expertise |
|
05. |
A clearer description of the results, with direct references to the studies analyzed, would be desirable. At the moment, in fact, the entire third paragraph is extremely chaotic and unstructured. |
Entire result section was restructured according to the three modified research questions. Respectively 3.1 Characteristics of the literature 3.2 Existing strategies for urban planning in tsunami-prone areas Existing strategies for urban planning in tsunami-prone areas
3.3 Urban planning in tsunami prone areas of Sri Lanka
|
|
06. |
Divide the discussion by the conclusions |
Divided in to two section as Discussion (Section 4) and Conclusion (Section 5) |

Round 2
Reviewer 3 Report
The quality of the paper has improved.
Minor spell check required.